# Water sources aggregate parasites with increasing effects in more arid conditions

Georgia Titcomb [1,2 ✉], John Naisikie Mantas[2], Jenna Hulke[3], Ivan Rodriguez[1], Douglas Branch[4] & Hillary Young [1,2]

Shifts in landscape heterogeneity and climate can influence animal movement in ways that profoundly alter disease transmission. Water sources that are foci of animal activity have great potential to promote disease transmission, but it is unknown how this varies across a range of hosts and climatic contexts. For fecal-oral parasites, water resources can aggregate many different hosts in small areas, concentrate infectious material, and function as disease hotspots. This may be exacerbated where water is scarce and for species requiring frequent water access. Working in an East African savanna, we show via experimental and observational methods that water sources increase the density of wild and domestic herbivore feces and thus, the concentration of fecal-oral parasites in the environment, by up to two orders of magnitude. We show that this effect is amplified in drier areas and drier periods, creating dynamic and heterogeneous disease landscapes across space and time. We also show that herbivore grazing behaviors that expose them to fecal-oral parasites often increase at water sources relative to background sites, increasing potential parasite transmission at these hotspots. Critically, this effect varies by herbivore species, with strongest effects for two animals of concern for conservation and development: elephants and cattle.

[1] Department of Ecology, Evolution and Marine Biology, University of California, Santa Barbara, CA, USA. [2] Mpala Research Centre, Laikipia, Kenya. [3] Department of Biology, Texas A&M University, College Station, TX, USA. [4] University of the West of England, Bristol, UK. ✉email: georgiatitcomb@gmail.com

Climate change is predicted to lead to dryland expansion over half of the globe's land by 2100[1], increasing the importance of surface water for wildlife, domestic animals, and the two billion people currently living in water-stressed areas[2]. However, water sources have great potential to serve as transmission foci for a range of diseases in a landscape, as they likely concentrate a wide range of hosts in a small area where parasite exposure may be increased[3,4]. In particular, for many parasites that can be transmitted via the environment, landscape heterogeneity can create localized transmission hotspots that have the potential to markedly affect overall parasite exposure risk[5–7].

Water sources can draw animals together, particularly in dry conditions[8–10]. For example, water sources drive large elephant aggregations[11] and increase contact rates among cattle herds[12] during dry periods. However, the degree to which different herbivores gather at water may vary by diet[13], physiology[14], and predation risk[14,15]. Camera trapping work of animal overlaps at watering holes and at baited food stations has shown species-specific increases in contact rates around resources[16–18] suggesting potential implications for disease transmission. However, there has been no large-scale experimental work measuring the degree to which animals congregate around water relative to their background density, how this may impact disease risk, or how this varies across climatic contexts. Understanding these patterns will provide critical new information about relative parasite risk across contexts and at relevant spatial scales[18]. This will be increasingly important given that water manipulation (e.g., changing waterholes, dams, and rivers) is widespread and increasing across the globe, particularly in water-limited landscapes.

Resources that attract hosts and increase contacts with other hosts, vectors, or infectious stages in the environment can act as hotspots of parasite risk. While surface water sources are established as hotspots for diseases with obligate water development of parasite or vector (e.g., mosquitoes that must develop in water sources), there is little understanding on the effects of water on density-dependent parasites transmitted via the fecal–oral route. Such parasites are likely to be particularly affected by host congregations around water, and include many medically and economically important gastrointestinal nematodes (order Strongylida)[19]. While parasites are important components of health-intact ecosystems[20], and are key to population regulation, many impose significant health threats. Gastrointestinal nematodes often inflict serious morbidity on domestic and wild herbivores (Supplementary Appendix Tables S1 and S2), and, while no study has quantified the global economic losses attributable to these worms, estimates in Europe show 10–50% production losses on farms due to these parasites[21] and growing worldwide concerns of rising resistance of these parasites to treatment[22]. Additionally, gastrointestinal worms cause disease in more than 2 billion people worldwide, particularly for those who are experiencing poverty[23]. Health impacts from nematodes can be exacerbated by additional stressors: for example, animals with low nutrition can have heavier infections[24] and those with reduced immune function can have increased mortality[25]. Thus, shifting host nutrition and immunity due to land use change[26], pollution[27], and climate changes[28] warrant more careful monitoring of parasite infections in wildlife across contexts.

Many gastrointestinal nematodes are transmitted via the fecal–oral route, releasing thousands of parasitic ova into the environment upon host defecation. These parasites develop in the environment before infecting hosts when they drink water or consume food contaminated with infective parasite stages from feces (e.g., strongylid nematodes[29]). While parasite egg density is often correlated with exposure risk, this pattern can vary if parasite survival varies significantly among sites. For instance, increased parasite egg density

near water might not translate to increased parasite exposure risk if reduced vegetation cover near water increases ground temperatures[30] or reduces moisture[31] such that larval mortality is increased; or conversely, if increased moisture from groundwater improves survival. In theory, increased time that hosts spend at water should lead to increased dung density (and thus parasites), followed by an increased risk of exposure to infective stages via drinking and eating, such that water may create a potentially important hotspot of gastrointestinal parasite risk in a landscape. However, this putative link between increased host activity and parasite exposure at water sources has only been supported by models[32] and an observational study on red deer[4]. In other systems, food resources have been manipulated to study effects on raccoon parasites[17], and carcass sharing among carnivores has been suggested to increase the potential for pathogen transmission[33]. However, there has been no large-scale experimental work testing the role of water sources in increasing the potential for parasite transmission.

While water-driven parasite aggregation likely occurs across a variety of landscapes where water is scarce and concentrated, East African tropical savannas provide an ideal place to investigate this phenomenon, as they are home to a diverse array of wild and domestic herbivores in a largely water-limited landscape. In addition to large numbers of ranched cattle (*Bos indicus* and *Bos taurus*), wild African herbivores include many locally or globally declining species such as zebra (*Equus quagga* and *Equus grevyi*), giraffe (*Giraffa camelopardalis*), elephant (*Loxodonta africana*), buffalo (*Syncerus caffer*), and impala (*Aepyceros melampus*), all of which are infected by a diverse community of helminths[34]. While many parasites are host-specific, substantial parasite sharing occurs even among taxonomically divergent species[35] when those species overlap spatially[36]. Notably, several important parasites (e.g., trichostrongyle nematodes; see Supplementary Appendix Tables S1 and S2 for a list of host-parasite records and pathogenic effects) are shared with closely-related domestic animals or with humans[34,37]. While many parasite-sharing links among different host species remain uncertain[37], several pose significant health threats to a range of hosts[38,39].

In this study, we first asked: Do water sources concentrate hosts, feces, and fecal–oral parasites across herbivore species? Implementing a two-year experiment in an East African tropical savanna (Fig. 1a, d), we expected that water removal and replenishment would respectively decrease and increase the density of water-dependent herbivores and their feces; and, based on parasite eggs in dung, parasite density in the environment. We expected results to be strongest for animals with high daily water requirements. Using additional data on herbivore grazing behavior, we then asked: Are water sources parasite exposure hotspots across herbivores? We combined herbivore grazing behaviors measured from camera traps with parasite density measurements to compare total potential exposures near water and matrix sites for four theoretical parasite mortality scenarios: equal mortality in the environment at water and matrix sites, and half, double, and ten times higher mortality rates near water. We expected that parasite density patterns would drive relative exposure results, with strongly aggregating herbivores showing elevated exposures near water, even with dramatically increased parasite mortality. Finally, we asked: How does the concentration of herbivore activity, herbivore dung, and parasite density at watering holes vary across rainfall contexts? Using observational data from water sources spread across a broad rainfall gradient (Fig. 1a) and over three years of sampling, we tested our hypothesis that herbivores, their dung, and parasite density would be more concentrated near water following periods of low rainfall and in more arid areas, and that effects would be strongest for highly water-dependent animals.

Here we show, using both experimental and observational methods, that parasite density is greatly elevated at water sources,

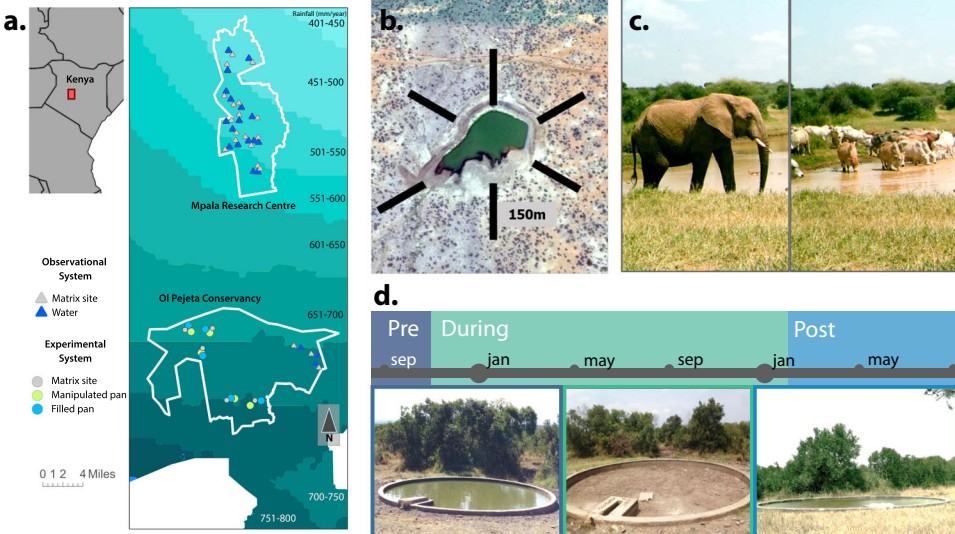

**Fig. 1 Experimental and observational study sites were located in central Kenya. a** We used five pairs of experimental water pans (blue and green dots) with matched matrix sites (gray dots) at Ol Pejeta Conservancy (OPC) and 20 pairs of observational dams (blue triangles) and matrix sites (gray triangles) across a rainfall gradient (teal shading, redrawn from data in ref. [72]) at OPC and Mpala. **b** Schema of sampling transects that radiated outwardly from both water pans and dams. **c** Sites were consistently utilized by both wildlife and domestic animals as measured by camera traps. **d** Experimental pans were filled and surveyed at the beginning of the study ("Pre", $n = 1$) before being drained ("During", $n = 5$) and refilled ("Post", $n = 3$). Image credits: **b** Source: "Mpala Research Centre", 0° 22′ 58.91″ N, 36° 51′ 33.77″ E. Google Earth. February 5th, 2015; accessed October 26th, 2021. **c, d** Photographs taken by study authors.

but that this varies by herbivore species. Notably, cattle and elephants drive variation in parasite eggs from dung, due to their high biomass, high infection intensity, and a high degree of aggregation at water (Fig. 2a–c and Supplementary Appendix Fig. S6). After considering herbivore behaviors and theoretical parasite mortality scenarios, this results in significantly elevated exposure risk for elephants and cattle in particular (Fig. 2d–f, and Supplementary Appendix Tables S13, S14). Water removal and replenishment changes herbivore, dung, and parasite aggregation (Fig. 3), and drier contexts amplify aggregations and parasite density near water (Fig. 4). Thus, when water availability is reduced—a global pattern that is increasing amid climate changes and growing anthropogenic water use—risk of parasite exposure may increase substantially if parasite mortality in the environment is not also greatly reduced. These findings are important for understanding shifting parasite dynamics for several threatened wildlife species and for pastoral livelihoods in response to changing water supply due to climate changes.

## Results

**Effects of water sources on hosts, dung, and fecal–oral parasites.** Water removal from five ~25,000 litre pans resulted in significantly reduced total, grazing, and drinking activity (measured via camera traps) at experimental water sources relative to filled water sources for elephants and cattle. While drinking activity was significantly reduced for all animals together, the 46% drop (i.e., a two-fold reduction) in total activity at experimental pans was only marginally significant ($t = −1.62$, $p = 0.1$ for the drained × during interaction, Supplementary Appendix Tables S7, S8, Fig. S4). Interestingly, total, grazing, and drinking activity increased when water was replenished such that it even trended higher compared to pre-experimental levels ($t = 2.27$, $p = 0.02$ for the drained × post interaction, Supplementary Appendix Tables S7 and S8). The interaction between experimental status and treatment was not an important parameter for models of buffalo, zebra, giraffe, or impala activity. However, after pans were refilled, zebra and buffalo activity was significantly higher at

experimental pans relative to other phases of the experiment ($p = 0.03$, $p = 0.01$ for total activity, $p = 0.09$ and $p < 0.001$ for grazing, $p = 0.01$ and $p < 0.001$ for zebra and buffalo respectively; Supplementary Appendix Tables S7 and S8).

Dung density at filled water sources relative to experimental water sources increased when water was drained for all animals together and herded cattle and elephants separately (Table 1 and Supplementary Appendix Table S9). This effect was largest for elephants: when experimental pans were drained, dung density was over six times higher at filled pans (in the area closest to water), while we did not find evidence of a difference pre-draining or post-refilling ($t = 2.88$, $p = 0.004$ for increased probability of zeros for the drained × during interaction, Table 1 and Supplementary Appendix Table S9). We found a similar pattern for cattle, as dung aggregation at filled pans was over three times higher during the drained period (at the 0 m mark), but not during the "pre" phase ($t = 3.09$, $p = 0.002$ for the increased probability of zeros for the drained × during interaction, Table 1 and Supplementary Appendix Table S9). While cattle dung density was significantly lower at drained pans relative to filled pans after refilling, this effect was substantially reduced. Cattle and elephants accounted for the largest proportion of dung density (>50%, Fig. 1b), driving a similar pattern for total dung. However, we did not detect a significant effect of water draining for zebra, impala, or giraffe considered separately, and buffalo dung density was slightly higher at experimental sites after refilling.

Total parasite density (order Strongylida)—estimated as the product of dung volume in the environment, dung physical density, and median fecal egg count for each species—was three times higher at filled pans compared to experimental pans during the experiment but was not significantly different before or after ($t = −2.93$, $p = 0.003$ for the drained × during interaction, Table 1, Fig. 3, see Supplementary Appendix Fig. S7 for species-specific responses).

Finally, parasite egg counts in dry soil near water pans were relatively low but consistent across treatments throughout the experiment ($X^2_2$ for status × treatment = 0.14, $p = 0.93$),

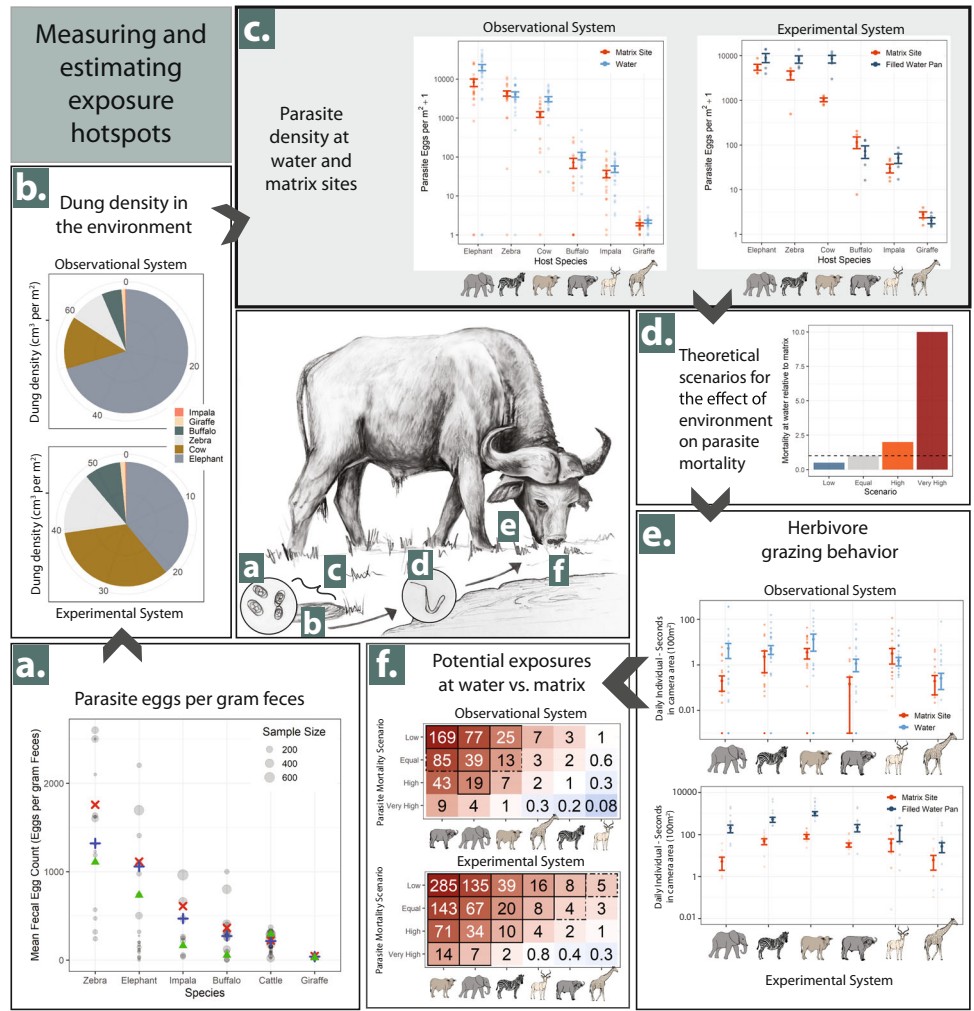

**Fig. 2 Measurements of parameters used to calculate parasite density in the environment and to estimate the degree to which potential parasite exposures are elevated near water relative to matrix sites. a–f** Key transmission steps noted in the central figure. **a** Fecal egg counts measured in this study (green triangles) compared to those reported in studies across the African continent (mean of individual studies in blue; studies weighted by sample size in red). **b** Average dung density contributed by each major species in both experimental (OPC) and observational systems (Mpala). **c** Comparisons of estimated parasite eggs contributed by each species at permanent water sources (either dams or pans, shown in blue) and matrix sites (calculated from both experimental and observational data sets, shown in orange) show considerable consistency across species. Estimates from the experimental system compare filled pans only to matrix sites. Note that both graphs are visualized on the $\log_{10}$ scale. Bars and centers represent means ± SE calculated from parasite density at the site level, averaged across all periods (observational system: $n = 20$ per species and treatment, experimental system: $n = 5$ per species and treatment). **d** Four theoretical scenarios of parasite mortality at water sources relative to matrix sites, ranging from an assumption of reduced mortality (low; in blue) due to potential increases in ground moisture, to an assumption of greatly increased mortality (very high; in red) due to decreases in vegetation cover. **e** Herbivore grazing activity at water sources (in blue) and matrix sites (in orange) for both experimental and observational systems. Note that both graphs are visualized on the $\log_{10}$ scale. Bars and centers represent means ± SE calculated from average daily grazing activity at the site level, averaged across all periods (observational system: $n = 12$ per species and treatment, experimental system: $n = 5$ per species and treatment). **f** Relative number of potential parasite exposures at water relative to matrix sites for each species and parasite mortality scenario from **d**. Significant ($p < 0.05$) and marginally significant ($p < 0.1$) differences for two-sided $t$-tests with Holm adjustment for multiple comparisons are bordered by solid and dotted lines respectively. Source data are provided as a Source Data file. Centre panel and icon artwork by G. Titcomb.

suggesting that water removal did not substantially affect dry soil parasite density. However, after including eggs found in wet soils, density was 16 times higher at filled pans during the experiment, but not significantly different before or after (based on dry weight, $X^2_2$ for status × treatment $= 9.30$, $p = 0.01$), since wet soils were not present at drained pans. Additionally, soil egg density in dry soils around both pan treatments averaged 4.5 times that at non-water sites ($X^2$ for treatment $= 5.58$, $p = 0.02$).

**Effects of water on potential parasite exposure**. Combining dung parasite density results (Fig. 2a–c) with herbivore grazing

behaviors showed that total potential parasite exposures for cattle and elephants were more than an order of magnitude higher near water if parasite mortality was equivalent at water and matrix sites (Fig. 2d–f). For the observational system, buffalos and elephants showed the strongest increase in total potential parasite exposures near water (85 and 39 times respectively, $p = 0.09$ and 0.002), with parasites of all other species except impala also trending higher. The strong results for buffalo were driven by the very low levels of buffalo observed grazing at matrix sites. With increasingly more conservative assumptions about parasite mortality differences across site types (assuming higher mortality of parasites near water), this effect diminished: at our most

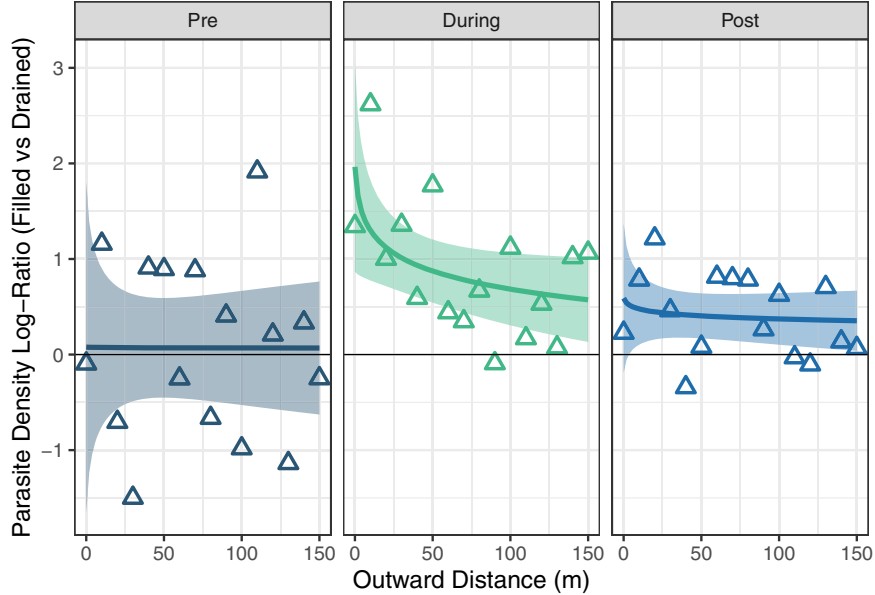

**Fig. 3 Log ratio of parasite density from dung at filled water pans relative to experimentally drained water pans throughout the experiment (pre-draining, during experiment, and post-refilling).** Points and lines that lie above 0 indicate increased density at filled pans relative to experimental pans. Points show parasite density log-ratios at each 10 m outward distance interval and experimental status (averaged over site and period); colors show each experimental phase. Lines represent best linear fits (±SE) to the points (using log(Distance)). Species-specific figures illustrating zero-inflated data are available in Supplementary Appendix Fig. S7. Source data are provided as a Source Data file.

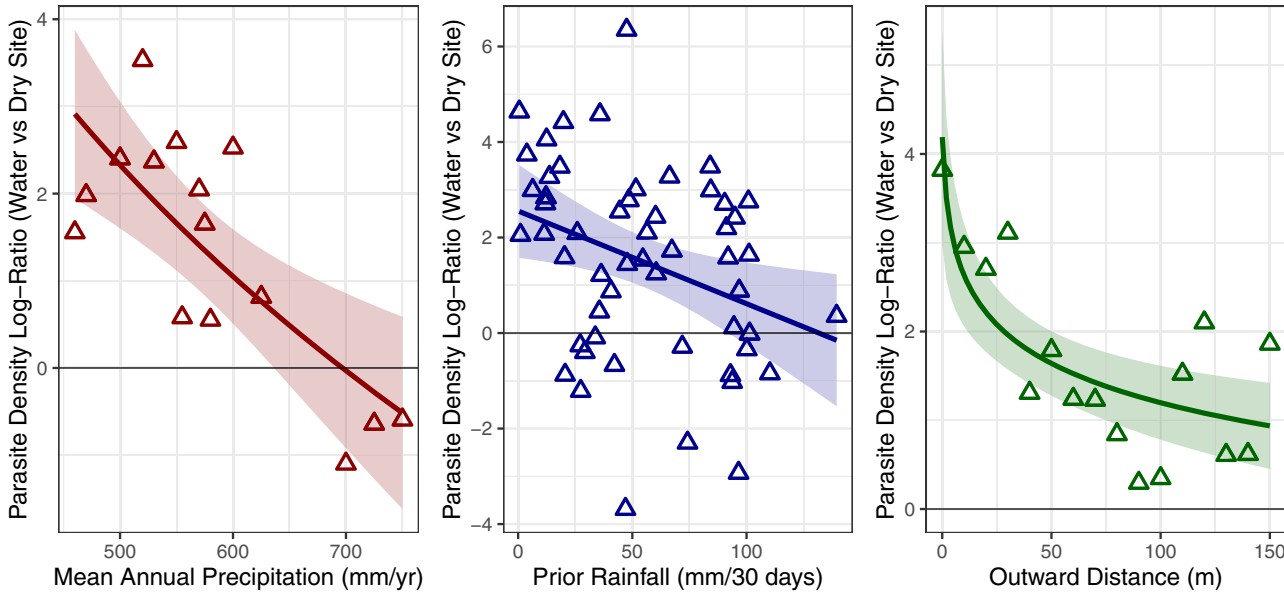

**Fig. 4 Visualized log ratio of parasite density from dung at watering holes relative to matrix sites across differing levels of mean annual precipitation, prior rainfall, and outward distance.** Points and lines that lie above 0 indicate increased density at water relative to matrix sites. Points represent averages for each value of MAP, prior rainfall, or outward distance; colors show different covariations with water limitation. Lines represent best linear fits (± SE) to the points. Species-specific figures illustrating zero-inflated data are available in Supplementary Appendix Fig. S8. Source data are provided as a Source Data file.

conservative assumptions (10× higher parasite mortality near water), only buffalo, elephants, and cattle had elevated potential parasite exposures at water, although this increase was not statistically significant (Fig. 2f).

For the experimental system comparisons between permanently filled water sources and matrix sites, potential parasite exposure was elevated near water for cattle, elephants, zebra, and impala (143, 67, 20, and 8 times higher, respectively, $p < 0.001$ for all but impala for which $p = 0.005$). This remained significantly elevated (14 times and 7 times higher) for cattle and elephants,

even in the scenario when parasite mortality between egg and infection was 10 times greater near water ($p < 0.001$, $p = 0.002$, respectively; Fig. 2f; all results reported in Supplementary Appendix Tables S13 and S14).

GIS analyses showed that the 150 m zone surrounding non-riparian water sources accounted for 1.54% and 2.61% of the total landcover at Mpala Research Centre and Ol Pejeta Conservancy, respectively (Supplementary Appendix Fig. S11). Weighting relative exposure ratios by these percentages showed that for several species, water sources had the potential to account for a

**Table 1 Significant fixed-effect coefficients for hurdle GLMM models of dung and parasite density for the experimental system are presented for both the conditional and zero-inflation components of the models ("Cond.", and "Zero").**

| Species | Model component | Status [during] | Status [post] | Treatment [drained] | Outward distance | Status [during]: treatment [drained] | Status [post]: treatment [drained] |
|---|---|---|---|---|---|---|---|
| All | Cond. | | | | − (<0.001) | − (0.002) | |
| | Zero | | | | + (<0.001) | | |
| Elephant | Cond. | | | | | | |
| | Zero | | | | + (<0.001) | + (0.004) | + (0.04) |
| Cow | Cond. | | | | − (<0.001) | | |
| | Zero | | | | + (<0.001) | + (0.002) | + (0.03) |
| Zebra | Cond. | | | | | | |
| | Zero | | | + (0.02) | − (<0.001) | | |
| Buffalo | Cond. | | | | | | |
| | Zero | | | | + (<0.001) | | − (0.02) |
| Impala | Cond. | − (0.03) | − (0.03) | | | | |
| | Zero | | | + (0.01) | − (0.03) | | |
| Giraffe | Cond. | | | | | | |
| | Zero | | | | − (0.02) | | |
| Parasites | Cond. | | | | − (<0.001) | − (0.003) | |
| | Zero | | | | + (<0.001) | | |

Parameters that predicted increasing and decreasing egg density are shown with a "+" and "−", respectively, for the conditional component, while parameters that predicted increasing and decreasing probability of a zero (i.e., no dung or parasites) are shown with a "+" and "−", respectively, for the zero-inflation component. Significant p-values for two-sided t-tests for each coefficient (unadjusted) are given in parentheses. The intercept corresponds to 0 m from water at filled pans prior to conducting the experiment ("Pre"). $N = 1440$ measurements per species. Full model results are available in Supplementary Appendix Table S9.

large proportion of parasite exposures across a landscape, especially at Ol Pejeta Conservancy. Under the assumption that matrix sites were representative of all landscape areas beyond the 150 m zone surrounding water, and that parasite mortality was equivalent at water and matrix sites, we found that water sources could account for up to 82% and 67% of exposures for cattle and elephants, respectively. However, these numbers were considerably higher than at Mpala Research Centre (17% and 38% for cattle and elephants, respectively). Effects were generally smaller for buffalo, giraffe, and impala (<20%), with the exception of buffalo at Mpala. Full results are reported in Supplementary Appendix Table S15.

**Effects of rainfall context on host and parasite aggregation.** Total herbivore grazing activity measured from camera traps was twice as high at water sources compared to matrix sites ($t = 2.51$, $p = 0.01$ for all species together), and it was significantly elevated for elephants, and marginally elevated for buffalo and zebra (Fig. 2e). The interaction between mean annual precipitation (MAP) and animal grazing at water was not significantly different from zero, although both predictors were important separately (Supplementary Appendix Table S8). This was likely due to the short deployment duration and low statistical power, as activity declined with increasing annual precipitation at both matrix sites and water sources, and it tended to be further elevated near water in drier areas (Supplementary Appendix Table S8 and Fig. S5). For all species except impala, grazing activity trended higher at water compared to matrix sites.

MAP and prior rainfall were important parameters in cattle, elephant, zebra, and total dung density models (Table 2 and Supplementary Appendix Table S10, Fig. S8). In dry locations (~460 mm/year) close to water and following no rainfall, cattle dung density was three orders of magnitude higher at water relative to matrix sites, but this elevated density decreased as MAP and outward distance increased. This pattern was also strong for elephants: in dry areas following periods of no rainfall, elephant dung was approximately ten times higher close to water, but this effect weakened as MAP and outward distance increased (Table 2 and Supplementary Appendix Table S10, Fig. S8). Zebra dung density did not differ significantly between water and

matrix sites when there was little prior rainfall or low MAP, and we even observed potential aversion to water during the wettest periods in high MAP areas. Impala dung density was also slightly elevated near water in low-rainfall locations but depressed near water in wet conditions. We observed slightly higher dung density levels at watering holes relative to matrix sites for buffalo and giraffe in low-rainfall conditions, and there was a significant interaction between MAP and prior rainfall for giraffe (Table 2 and Supplementary Appendix Table S10, Fig. S8).

Critically, outward distance from water, MAP, and prior rainfall all modulated parasite density at water sources compared to matrix sites. In areas where MAP was lowest (450 mm/year) and prior 30-days rainfall was 0 mm, parasite egg density from dung was estimated to be more than 150 times higher than matrix sites in the closest area to water. This effect decreased sharply as MAP, prior rainfall and outward distance increased (MAP: $t = 3.40$, $p = 0.001$; prior rainfall: $t = 5.06$, $p < 0.001$; distance: $t = 3.01$, $p = 0.003$, Table 2 and Supplementary Appendix Table S10, Fig. 4).

In our negative binomial model of eggs found in soil, parasite densities differed significantly based on water proximity ($X^2_3 = 109.73$, $p < 0.001$). Parasites (eggs per 20 g dry soil) were approximately two orders of magnitude higher in damp soil (based on dry weight, mean ± SE = 31.6 ± 11.5), and four times higher in dry soil (0.90 ± 0.31), near the water's edge compared to locations 1 km from water (0.23 ± 0.09) (Supplementary Appendix Fig. S9). There was a marginally significant effect of rainfall ($X^2_1 = 2.90$, $p = 0.09$) in which eggs were more abundant in wetter locations; however, this was consistent across sampling locations near and far from water ($X^2_3 = 2.68$, $p = 0.44$ for the interaction between sample type and MAP; Supplementary Appendix Fig. S10).

## Discussion

Utilizing experimental and observational data sets, we show that water sources strongly concentrate herbivores, herbivore dung and parasite egg density, with increasing effects in dry conditions with low prior rainfall and low MAP. Effects were greatest for cattle and elephants which drove patterns in total dung and estimated parasite density due to the high relative abundance of

**Table 2 Significant fixed-effect coefficients for hurdle GLMM models of dung and parasite density for the observational system are presented for the conditional and zero-inflation model ("Cond" and "Zero").**

| Species | Model component | MAP | Site type [water] | Distance | Rain | MAP: site type [water] | Distance: site type [water] | Rain: site type [water] |
|---|---|---|---|---|---|---|---|---|
| All | Cond. | | + (<0.001) | | − (<0.001) | − (0.05) | − (<0.001) | |
| | Zero | + (<0.001) | − (<0.001) | | + (<0.001) | + (0.002) | + (0.01) | + (<0.001) |
| Elephant | Cond. | | + (0.001) | | | | − (<0.001) | |
| | Zero | + (0.002) | − (<0.001) | | + (<0.001) | | + (<0.001) | |
| Cow | Cond. | | + (0.009) | | | − (0.03) | | |
| | Zero | + (0.003) | − (<0.001) | − (0.03) | + (0.001) | | + (<0.001) | |
| Zebra | Cond. | | | | | | | |
| | Zero | | | | + (0.02) | | | + (0.004) |
| Buffalo | Cond. | | | | | | | |
| | Zero | − (<0.001) | − (0.005) | | + (<0.001) | | | |
| Impala | Cond. | | | | − (0.03) | | | |
| | Zero | | − (0.003) | | | | | + (0.002) |
| Giraffe | Cond. | | | | | | | |
| | Zero | | − (0.04) | | | + (0.01) | | + (0.01) |
| Parasites | Cond. | − (0.002) | + (<0.001) | | − (<0.001) | − (0.03) | − (<0.001) | |
| | Zero | + (<0.001) | − (<0.001) | | + (<0.001) | + (0.001) | + (0.003) | + (<0.001) |

Parameters that predicted increasing and decreasing egg density are shown with a "+" and "−", respectively, for the conditional component, while parameters that predicted increasing and decreasing probability of a zero (i.e., no dung or parasites) are shown with a "+" and "−", respectively, for the zero-inflation component. Significant p-values for two-sided t-tests for each coefficient (unadjusted) are given in parentheses. The intercept corresponds to dung and parasite density at matrix sites when distance and prior rainfall are zero and MAP is the lowest level observed (450 mm/yr.). N = 2816 measurements per species. Full model results are available in Supplementary Appendix Table S10.

dung and average parasite fecal egg count. However, all species showed at least some negative interaction between dung density at water sources and annual or recent rainfall, suggesting that fecal–oral parasite density at water sources will be elevated in drier conditions for all species.

**Effects of water sources on hosts, dung, and fecal–oral parasites.** Experimental findings showing strong reductions in total dung density and estimated parasite density with water removal were driven by two globally important species: cattle and elephants. Weaker responses for other herbivores likely demonstrate differing balances of resource requirements, predation risk, and parasite exposure risk.

Both elephants and cattle are highly water dependent[40,41] compared to several other animals in our study. Elephants can quickly alter their movements in response to water availability[42]. Cattle, whose movements are typically dictated by humans, reflect the rapid ability for humans and their livestock to adapt to changes in water distribution. In contrast, lower water removal responses for other herbivores may be explained by foraging and water-seeking tradeoffs. If forage quantity or quality is reduced near water, some species may drink and depart, rather than stay and forage near water[43]. In another study in this system, understory height, grass, and forb cover were reduced near water[44], consistent with findings in other systems[45]. For large water-dependent grazers, such as zebra and buffalo, foraging requirements might limit their ability to consume sufficient material near water[43]. Browsers, such as giraffes and impala are less water-dependent as their digestive systems allow for better water retention[46]. Lack of suitable browse near heavily impacted water sources may explain why giraffe dung is not substantially elevated at water. Finally, species-level differences in defecation behavior may also explain observed variation. For example, impala tend to defecate in middens, which have been shown to increase the density of surrounding infective larvae themselves[47]. If middens are independent of water location, then dung density is unlikely to tightly correlate with water proximity.

Elevated predation risk at water sources[13,48] may also explain lower aggregation levels for certain herbivores. This explanation is parsimonious with our results given that the large body size of elephants affords some protection from predation[49] and in this system cattle move exclusively with human protection, reducing both predation risk and the ability of animals to respond to that risk. Heightened risk for smaller herbivores[49] may explain why impala, which are strongly constrained by predators[50], had little dung accumulation near water, despite having moderate water requirements[43]. Indeed, our camera trapping data showed more than twice as much carnivore activity at water sources at both Mpala Research Centre and Ol Pejeta Conservancy than at matrix sites. It thus seems likely that predators may indirectly influence herbivore fecal–oral parasite exposure via a landscape of fear.

**Effects of water on potential parasite exposure.** Our results suggest that water sources are hotspots of parasite exposure (and thus, transmission), especially for parasites of highly water-dependent species, such as elephants and cattle. While this corresponded to increased parasite egg density patterns, it was also influenced by increased herbivore grazing behavior near water (Supplementary Tables S7 and S8). Indeed, despite several studies documenting severe vegetation loss around water in highly arid conditions[51], water sources can also promote grazing lawns attractive to herbivores that may increase parasite exposure[44]. Furthermore, our results show that microclimatic conditions at water sources would need to induce at least 10 times higher parasite mortality for total transmissions to be only modestly elevated compared to matrix sites for highly water-dependent species (Fig. 2F and Supplementary Table S15 and S16). This may be unlikely, given that nematodes often migrate through soil to optimize the balance between increasing survival and increasing transmission probability, and can often avoid severe surface conditions[30]. Indeed, moist conditions near water may mitigate parasite desiccation. Consistent with this, our finding that damp soil, as opposed to dry soil, had higher egg density could reflect increased input from dung, increased egg survival, and/or delayed hatching. The relatively low parasite egg density and lack of treatment effect in dry soils indicates that larvae likely develop and disperse into soil and surrounding vegetation quickly, where their survival and probability of infecting a host also likely varies by parasite species.

In addition to microclimate influences on parasites, broader climate changes will affect parasite survival in the environment to varying degrees depending on parasite-specific resilience to changing temperatures and ability to adapt[52]. Specific parameters are unknown for most of the wildlife parasites in this system, but future modeling work may be able to explore the extent to which variation in survival, development, and dispersal will allow for more nuanced predictions across parasites and hosts, in addition to extensions for a wide variety of other pathogens transmitted via the fecal–oral route, including enteric viruses, intestinal protozoa, and bacteria. This is especially true given that our relative exposure calculations accounted for transmission via grazing only; for many of these additional pathogens that also spread via drinking, water sources will further amplify parasite exposure.

We found that the area immediately surrounding water could account for a large proportion of parasite exposures via grazing for elephants and cattle, despite comprising only a small fraction of the landscape. These findings rely on the assumption that areas more than 150 m from water are similar to matrix sites more than 1 km away, and they exclude other potential sources of exposure heterogeneity, such as cattle corrals and glades where many animals graze. However, exposure at these locations could be quantified in much the same way as this study to achieve more nuanced estimates of potential exposure across heterogeneous landscapes. Future work could assess the relative contribution of different types of hotspots to parasite transmission, as demonstrated previously in the case of anthrax transmission, while also accounting for their landcover, as proposed in ref. [5].

While these study results have clear implications for parasitism within individual herbivore species, the consequences of the total increase in parasite density on other species will depend on interspecific parasite sharing. While parasite identifications were not possible from morphological analysis, literature reviews indicate that all hosts share at least one parasite with at least one other host species[34]. However, elephants rarely share nematode parasites with other species[34], indicating that the strong implications of water in concentrating elephant parasites are likely largely confined to elephants. However, cattle share multiple gastrointestinal helminths with many species[37], and their strong aggregations at water may increase parasite exposure for other nearby foraging or drinking animals. Given that cattle comprise approximately one-half of all herbivore biomass in the broader region[53], even a small degree of overlap in parasite sharing with other host species may substantially affect parasitism in other wildlife[18]. This may be another way in which human domination of landscapes increases threats to wildlife. However, cattle anthelminthic treatment has the promise to reduce shared helminth transmission, and could alleviate parasite sharing risk in areas where this is of conservation concern, although increasing anthelminthic resistance is a growing issue[22].

**Effects of rainfall context on host and parasite aggregation.** Observational results expand upon experimental findings, showing that lower recent rainfall and MAP can further concentrate animal dung around water. While effects varied by species, elephant, cattle, giraffe, and buffalo dung were concentrated more strongly at water sources in areas of low MAP, suggesting that the density of their fecal–oral parasites increases when water is limiting. Furthermore, several species were dependent on short-term rainfall—during dry periods (<50 mm rain over 30 days), giraffe, buffalo, zebra, and impala dung were elevated at water, suggesting a shifting tradeoff between water requirements and forage and/or predation risk. Indeed, the context of water limitations may explain why these animals demonstrated lower (or no) aggregation levels at our experimental site, which was located in an area

of higher annual rainfall (Fig. 1a). Together, these results suggest that climatic context can alter fecal–oral parasite dynamics at water sources for these species. While future annual rainfall projections in our study region are mixed[54], local models predict seasonal long rain reductions[55]. Globally, increased temperatures[56], broadscale aridification[1], and increased competition for water with humans[57] may drive certain wildlife to congregate more strongly at water, likely increasing fecal–oral parasite exposure.

Costs of certain nematode infections for many host species are non-trivial. For instance, nematode infections in elephants—a species classified as vulnerable by the IUCN—are common (Supplementary Appendix Table S2), and at least one study identified parasitism and starvation as likely causes of death for 38 young elephants that died during a period of severe drought in Kenya[58]. This underscores the importance of incorporating likely changes in parasite infection risk as part of conservation planning for the management of this species amid changing environmental conditions. Large strongyles that infect equids, including endangered Grevy's zebra, can cause severe effects, including colic and death, and while nematode pathology in wildlife is not well understood, a wide array of trichostrongyles and hookworms cause substantial economic losses for livestock[59] (Supplementary Appendix Tables S1 and S2).

In addition to direct impacts on animal health, parasite aggregation may have indirect impacts on animal behavior and fitness, much as predators may impose fitness costs to prey through the "landscape of fear" that, at the population level, often exceed the fitness costs of direct predation. Recent work has focused on the ability for "landscapes of disgust" to facilitate host avoidance of parasites[60,61]. Parasite avoidance behavior may complicate our findings if animals can detect parasites in water or the environment. For example, other studies have shown fecal avoidance in feeding dik-dik (a small antelope)[47] and elephants and lemurs seeking water[62,63]. These studies suggest that in certain cases, the costs of parasite exposure may alter animal behavior and foraging.

Notably, several study conclusions are based on average fecal egg count (FECs) measurements to estimate fecal–oral parasite density and transmission opportunities. While this assumption allowed us to ask questions at a larger scale, seasonality can affect FECs for different host and parasite species in different directions, and many studies have shown seasonal variation in herbivore FEC[24,64,65]. Any consistent seasonal deviation in infection intensity could dampen or heighten the effect of water. Studies have found both increases and decreases in FECs over rainfall seasons and droughts for the herbivore species examined in this study. However, previous work from this study site found that drought was associated with increased FECs for six of nine bovid species[24], with no species showing decreases. Thus, our results may be a conservative estimate of the impacts of drought on parasite egg aggregation in the environment, but this may be offset by increases in parasite mortality due to desiccation. These nuances underscore the importance of future work investigating combined host and parasite responses to ongoing climate changes.

This study shows that water sources cause large scale—up to 150-fold—increases in helminth parasites that are expelled into the environment compared to background density during the driest conditions, although the effects vary strongly across herbivore species. Accounting for herbivore grazing behavior and parasite survival scenarios revealed that this increase in parasite density translates to increases in total potential parasite exposures near water, except in severe circumstances when parasite mortality is more than an order of magnitude higher at the water and for less water-dependent species. Critically, we

show that climatic context greatly modifies patterns of parasite density, with stronger levels of parasite concentration in rainfall-limited times and locations. This suggests that water management—for both human and domestic animal use—is an important way in which humans influence wildlife parasite dynamics. This influence will likely only increase as water becomes increasingly scarce and livestock biomass continues to increase regionally[53] and globally[66]. Cumulatively, these findings highlight multiple potential pathways in which humans can affect wildlife parasitism and behavior via climate change and domestic animal management.

## Methods
Research was conducted at two mixed wildlife and cattle ranching properties in Laikipia county, central Kenya: Ol Pejeta Conservancy and Mpala Research Centre (Fig. 1).

**Experimental system**. At Ol Pejeta Conservancy (0.0043° S, 36.9637° E), we established five experimental sites, each with one pair of water pans (10 pans total) and 1 "dry" (no-pan) site (Fig. 1a). Matrix site coordinates were randomly selected from a range of locations 1 km from the experimental water pan and at least 1 km from any other water source. One of the two water pans (located 0.4–1 km apart at each site) was drained (experimental pan) for one year and then refilled, while one remained filled throughout the experiment (filled pan) (Fig. 1b, c). We measured herbivore activity via camera traps and dung surveys. Camera traps were used to verify elevated herbivore activity patterns suggested by dung density, and to measure the density of grazing behaviors that would expose herbivores to parasites. Cameras ran for 5856 trap nights from August 2016–September 2018. Full camera trapping methods are provided in the Supplementary Appendix Figs. S1, S2 and Table S3. We performed dung surveys at each site once before draining water from each experimental pan in October 2016. We repeated dung surveys at each pan and matrix site (every 3 months, $n = 5$ surveys during the experiment) before refilling in January 2018 and resurveying ($n = 3$ surveys post refill). Surveys were performed along six 150 m transects extending radially outward from the water source (or matrix site center). A 1 m$^2$ quadrat was placed every 10 m ($n = 16$ quadrats per transect), and volume of all large mammalian herbivore dung was estimated and classified as "fresh" (≤3 days) or "old" (>3 days) (see Supplementary Appendix, Fig. S3 and Table S5 for detailed dung volume measurement methods). After a drought in June 2017, quadrats were laid on both sides of the transect to increase the sampling area, and density was calculated by averaging across the two quadrats.

**Observational system**. To investigate herbivore abundance, dung density, and parasite density across climatic conditions, we extended sampling protocols to an additional 20 man-made dams (and associated matrix sites) at Mpala Research Centre (0.283° N, 37.867° E) ($n = 17$, described in ref. [44]) and Ol Pejeta Conservancy ($n = 3$). While experimental pans were confined to one rainfall zone (~700 mm/yr.), these additional 20 water sources spanned a 460–760 mm/year rainfall gradient (Fig. 1d), marking a transition from sub-desert scrub to grass/tree savanna[67]. We included paired matrix sites 1 km from the dam and at least 1 km from any other water source. From April to September 2017, one camera was placed at each dam and matrix site for one month (429 trap nights) (full methods provided in Supplementary Appendix and Table S4). Dung surveys were conducted using experimental system methods, except that quadrats were laid on only one side of the transect. Five surveys were conducted from November 2015 to October 2017 at all sites at Mpala; two surveys were repeated at the Ol Pejeta Dams during November 2015 and September 2016.

**Photograph analysis**. We uploaded camera trap photographs to a citizen science website (https://www.zooniverse.org/projects/gtitcomb/parasite-safari) where volunteers assisted in classifying images by counting animals that were present, drinking, and/or grazing. Image processing methods and validation are reported in detail in Supplementary Appendix Figs. S1 and S2. Briefly, images were aggregated into distinct trigger events if less than 5 min elapsed between images of the same species at a given location. We multiplied the mean number individuals present by the elapsed time of each trigger to estimate total animal activity (in units of individual-seconds; Cf. person-hours). We repeated this for the mean number of animals grazing to calculate the density of potential exposure events.

**Parasite detection**. We estimated parasite eggs in the environment (eggs/m$^2$) as the product of median fecal egg counts (eggs/g) by the physical density of fresh dung for each species (g feces/cm$^3$) and the volume of fresh dung in the environment (cm$^3$/m$^2$). Dung density calculations are detailed in Supplementary Appendix and Table S5. To quantify average parasite density, we conducted fecal egg counts (FECs) on fresh herbivore dung samples ($n = 131$) collected across multiple years and locations at Mpala (Fig. 1b and Appendix 4) using the mini-FLOTAC protocol (4 g feces in Sheather's sugar solution)[68]. To contextualize FECs,

we conducted a literature search of reported FEC values for focal herbivores. Our values fell within ranges reported throughout East and Southern Africa (Fig. 1B and Supplementary Appendix and Table S6).

We also measured parasite eggs at all sites by subsampling surface soil (< 1 cm depth) from the water's edge (0 m wet), dry soil next to the water (0 m dry), and 50 m from the center of the matrix site (1 km dry). We sieved (2 mm grain) and filtered 4 g soil from each of five transects to create a homogenized 20 g composite sample ($n = 175$ at Ol Pejeta; $n = 232$ at Mpala). For wet soils, we combined 5 g from five locations (25 g total) and calculated dry weight using a replicate composite sample. We followed a sedimentation-floatation protocol[69], using 0.1% Tween 80 to wash soil, and Sheather's sugar as a floatation solution to isolate eggs. We counted all unhatched and intact strongyle-type eggs that rose to the coverslip following 15 min of centrifugation at $500 \times g$. For wet soil samples, we used dry soil weight to calculate dry soil eggs per gram.

### Analyses
*Effects of water sources on hosts, dung, and fecal–oral parasites*

## Herbivore activity
We compared total, grazing, and drinking activity (daily individual-seconds) at filled and experimental pans recorded from camera traps using GLMMs with either a negative binomial or Tweedie error structure. The appropriate error structure was determined using both AICc comparisons and residual diagnostics using the DHARMa package[70]. We tested for significance of the interaction between experiment status (pre, during, or post) and treatment (filled or drained) for each herbivore species using $X^2$ tests. Site ($n = 5$) and month ($n = 7$) were included as random effects.

## Dung and parasite density
We compared parasite and dung density (eggs/m$^2$ and cm$^3$/m$^2$, respectively) at filled and experimental pans using generalized linear mixed hurdle models with zero-inflation and Gaussian conditional components. Density was cube-root transformed to meet residual assumptions (determined using the DHARMa package) for all models (elephants, cattle, zebra, giraffe, and buffalo), except for impala, which was log-transformed. Zebra dung densities reflect both *Equus grevyi* and *Equus quagga*, as their dung is indistinguishable. We tested the effect of experiment status (pre, during, and post) on differences between dung density at filled and experimental pans, assuming a significant interaction between status and treatment in either the conditional or zero-inflated components of the model signified changes due to water manipulation. We also included outward distance from water (log-transformed) as a fixed effect, while period ($n = 10$) and site ($n = 5$) were random effects. We also analyzed the log ratio of dung density for all dung and parasites summed together as exponentiating the log ratio provides an intuitive estimate of relative dung and parasite density. These models follow a similar structure as negative binomial GLMMs and have qualitatively similar results; they are presented in full in Supplementary Appendix Tables S11–S13.

We also compared dung density between matrix sites and filled pans to explore the possibility for underlying variation in the system as a driver of patterns seen. While dung density for most species differed at water sources compared to matrix sites, and was more than eight times higher (at the 0 m mark) summed across species, only impala dung density changed at filled pans compared to matrix sites throughout the experimental period, indicating that in almost all cases, significant results were likely a result of changes to experimental pans only (Supplementary Appendix Table S14).

## Parasites in soil
We used a negative binomial generalized linear mixed model to test whether soil parasite egg densities in (1) dry soil next to drained and filled water pans and (2) wet and dry soils together differed by experiment status. We used another negative binomial GLMM to test whether dry soil parasite density differed between water pans and non-water matrix sites. We included site ($n = 5$) and period ($n = 10$) as random effects in all models.

*Effects of water on potential parasite exposure*

## Total potential exposures
For parasites with infectious stages in the environment, potential parasite exposures per unit time and area can be thought of as the product of (a) parasite density in the environment, (b) the rate at which each host consumes parasites, and (c) the density of hosts in the environment[71]. We combined data on (a) parasite density from herbivore dung with (b and c) the product of elapsed time and mean count of individuals grazing (i.e., herbivore grazing activity in units of individual-seconds from Q1) to estimate relative parasite exposures per unit time and area for each herbivore species at permanently filled water pans and matrix sites for each of the five locations. Importantly, we assumed that (1) time spent grazing was proportional to parasite consumption rate, (2) the proportion of susceptible individuals was consistent at water vs. matrix sites, and (3) transmission dynamics were density-dependent. We then combined this ratio with four

different parasite mortality scenarios to estimate the potential exposure ratio:

$$\text{Potential exposure ratio} = \frac{H_w \times P_w}{H_m \times P_m} \times M \qquad (1)$$

Where $H$ is the density of grazing behaviors (individual-seconds per 100 m$^2$ covered by each camera with a 50-degree angle and 15 m detection distance) at water ($w$) or matrix ($m$), $P$ is the density of eggs in the environment from dung, and $M$ is the ratio of parasite mortality at matrix sites relative to water. We calculated the potential exposure ratio for low ($M = 2$), equal ($M = 1$), high ($M = 0.5$) or very high ($M = 0.1$) theoretical mortality scenarios near water relative to matrix sites. We used a GLMM with a Tweedie error structure to model the ratio of total potential parasite exposures using herbivore species and site type (water source or matrix) and their interaction as fixed effects, and location ($n = 5$) as a random effect. We performed post-hoc tests of pairwise differences between water sources and matrix sites for each species, using the Holm method to control for multiple comparisons ($n = 6$).

We also computed total potential exposures using the same methods for the observational system by combining parasite density data with herbivore grazing behavior and four different parasite mortality scenarios for each of 12 different locations with sufficient camera trapping data. We used a GLMM with a Tweedie error structure to model potential exposure ratio using herbivore species, site type (water source or matrix) and their interaction as fixed effects, and location ($n = 12$) as a random effect, again performing post-hoc tests of pairwise differences using the Holm method to control for multiple comparisons ($n = 6$).

Finally, to place these results in the context of the total landcover of our study sites, we quantified the percentage of land that fell within 150 m of a water source (excluding rivers and drainage areas), relative to the entire study area using ArcGIS Pro (v 2.5). We then weighted potential exposure ratios by this percentage to determine the ratio of total parasite exposures near water relative to the rest of the landscape, making the important assumption that potential exposures in all other landscape areas resembled values calculated from matrix sites (Supplementary Appendix Sect. 7 and Fig. S11).

*Effects of rainfall context on host and parasite aggregation.* To understand how rainfall impacted herbivore activity, dung, and parasite density, we used camera trap, dung count, and parasite data collected at paired water and matrix sites for the same species as in our experimental analyses.

## Herbivore activity

We compared total (i.e., animal presence, regardless of behavior) and grazing activity (daily individual-seconds) for all herbivores at water sources and matrix sites, again using GLMMs with either a negative binomial or Tweedie error structure. We tested for interactions between site type (water or dry) and MAP and prior 30-days rainfall, including random effects for the site ($n = 17$). MAP values were derived from ref. [72], and prior rainfall data were available from Mpala's long-term rainfall monitoring project[73].

## Dung and parasite density

To understand differences in dung and parasite density at water sources compared to matrix sites, we again used zero-inflated Gaussian hurdle models with a cube-root transformation to test interactions between water presence and prior 30-day rainfall, MAP, and outward distance, including random effects for site ($n = 20$) and period ($n = 5$). We again analyzed the log ratio of dung density and parasites summed together, which provided qualitatively similar results (Supplementary Appendix Table S11).

## Parasites in soil

Finally, we used a negative binomial GLMM to test whether soil parasite egg densities differed among sample type (wet soil, dry soil next to the water's edge, and dry soil 1 km from water), across a rainfall gradient, and whether there was an interaction between sample type and rainfall gradient. We included location ($n = 20$) and period ($n = 5$) as random effects.

All analyses were performed in R 4.0.1[74].

**Reporting summary**. Further information on research design is available in the Nature Research Reporting Summary linked to this article.

## Data availability

All camera trapping, dung density, and parasite density data generated in this study have been deposited in the Environmental Data Initiative (EDI) database, available publicly at: https://doi.org/10.6073/pasta/2728d61f10b767814b5d95fbd69137fa[75]. The data package also includes additional source data files for figures. Source data are provided with this paper.

## Code availability

Code for analyses is also available alongside data sets in the EDI package cited above (https://doi.org/10.6073/pasta/2728d61f10b767814b5d95fbd69137fa), in addition to https://github.com/gtitcomb/parasites_water_sources.

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

## Acknowledgements

This study was conducted on land originally occupied by different indigenous people including pastoralists, hunter–gatherer, and earlier human communities. During British colonial rule and continued adjudication in independent Kenya, land in Laikipia was converted to commercial and group ranches, communal lands, and conservation areas. Mpala's land and resources are managed by a board of trustees including international and Kenyan institutions focused on a mission of science, education, conservation, and community outreach. We thank Ol Pejeta Conservancy, Mpala Research Centre, and Kenya Wildlife Service for facilitating this work. Fieldwork for this project is permitted under the Kenyan National Commission for Science, Technology, and Innovation (NACOSTI/P/16/0782/10585) and Kenya Wildlife Service (KWS/BRM/5001). GCT was supported by the National Science Foundation Graduate Research Fellowship (1650114) and the National Geographic Society Early Career grant (EC-33R-18). This work was also supported by NSF DEB 1556786 awarded to HSY. We are grateful to Richard VanAardt, James Ngeso, and Benard Gituku for facilitating work at Ol Pejeta, and to Michelle Long, Edward Trout, and Valerie Lensch for additional field assistance. We are especially thankful to the many thousands of citizen science volunteers on the Zooniverse platform. Finally, we give our sincere thanks to Vanessa Ezenwa for feedback that improved this manuscript.

## Author contributions

G.T. designed the study, conducted field and lab work, performed analyses, and wrote the paper. J.M., J.H., I.R., and D.B. conducted field and lab work and provided feedback on the paper. H.Y. assisted with study design and writing the paper. All authors approved the final manuscript.

## Competing interests

The authors declare no competing interests.
