## [Peer Review File · Nature Communications]

Reviewers' Comments:

Reviewer #1:

Remarks to the Author:

This is a very interesting study that makes important contributions relevant for basic and applied research in disease ecology and conservation biology, and in addition, could be relevant for veterinary and public health. The results clearly demonstrate increased potential for the transmission of gastrointestinal parasites close to permanent water sources in dry areas. A central claim of the paper is that this increased transmission makes these water sources to hotspots of disease exposure. Unfortunately, I am not entirely convinced that the conducted analyses are sufficient for making this claim. However, I think the authors should be able to extend their analyses to fix this problem.

To assess whether water sources act as exposure hotspots the authors compare data collected directly at water sources and data collected at matrix sites with substantial distance to permanent water sources. In their analysis the authors correctly consider variation in parasites (including potential variation in their viability) and variation in individual grazing behaviour and the total time animals spend at either kind of site. A critical issue that seems to be ignored is the landscape-level distribution of areas close to water vs. areas further away from water. Specifically, in savanna ecosystems, areas close to water are usually only a small proportion of the whole landscape. Thus, even if there is increased exposure close to water, animals might spend so little time there that areas close to water still do not act as transmission hotspots. As an extreme example, if the exposure risk close to water is 10 times higher but animals spend only 1% of their grazing time in such areas then water sources would not really act as exposure hotspots. Or at least transmission close to water would not be of key importance for the overall transmission dynamics. Therefore, I think it would be very valuable if the authors would extend their analyses to estimate how much of the overall exposure across the whole landscape occurs close to water. I realize that the authors do not have the ideal data set to perform such estimations. Nevertheless, some rough estimates should be possible, which I think would be very valuable for judging the ecological relevance of parasite transmission close to permanent water sources.

In the discussion, the authors briefly discuss the potential for parasite transmission among species. It seems to me there is still some unused potential to explore this topic quantitatively using the collected data set. It might be possible to compare between-species transmission patterns (maybe in form of networks) between water and matrix sites. Such an analysis might reveal that parasite exposure between these sites does not only differ quantitatively (as already shown) but also to some degree qualitatively (in terms of who transmits to whom). I think this could be an interesting addition to the already performed analyses. However, I do not insist to include such an analysis in the current manuscript.

Minor comments:

I found the description of the aims in the last paragraph of the introduction not very clear, and more detailed explanations might help a lot. When reading this first I encountered some difficulties:

- There seems to be a contradiction between (1) the first questions which focuses on the quantitative effects of water and (2) the subsequent expectation that focuses on the qualitative effects of experimental water manipulation.
- For the second question initially, I did not understand the idea of the mortality scenarios. Here it would really help to spell out that you were able to measure this and instead assumed several theoretical scenarios in order to obtain exposure estimates.

The description of the analyses in the methods appeared to me partly not very well structured. At least I had sometimes difficulties imagining where each specific analysis is fitting into the bigger picture. Here it might be helpful to add information to which question each analysis belongs to. Alternatively, the methods might be easier to follow if the main structure would follow the questions instead of having the main division between experimental and observation parts.

The section on "total transmission" (L226 ff) could benefit from a more detailed description of what you are doing here. Specifically, it would be useful to explain what "total parasite exposures for

each herbivore species" means. As far as I understand it, this combines the exposure of individuals and the density of individuals. I first thought there is something wrong because I had expected that exposure should be analysed on the level of individuals. Only after some time, I realized that this is all correct because with the density of individuals you control for the effect that individuals aggregate at water and thus have a higher exposure risk there. It would be nice if this would be spelt out directly.

Throughout the results, statistically non-significant effects should not be reported as indications of the absence of an effect, e.g. as done in lines 310 and 356.

L388: is it really true that elephants and cattle are much more water-dependent than buffaloes? I thought their water dependence was comparable, but maybe I misremembered that.

Figure 2: I generally like this figure, I especially like the representation of the panels in the drawing in the centre. I still suggest emphasizing more clearly that the data in panel D do not depict real measurements and instead are assumptions that you put into the analysis. Maybe you could change the text to "Theoretical scenarios for the effect of.."

Reviewer #2:

Remarks to the Author:

I reviewed a previous version of this manuscript for a different journal. I made several suggestions that I am happy to see the authors addressed here by adding the second main goal, to investigate "to what extent are water sources parasite exposure hot spots across herbivores." Making the connection from numbers of parasite propagules in the environment to how host exposure risk varies based on heterogeneity in parasite survival under different environmental contexts is a significant strengthening of this study, and this will be an important contribution to the field. I do think, however, that the analyses for this part need more clarification for this connection to be clear.

What exactly is an "exposure" as defined in the data extraction/analysis? I struggled to understand this throughout, and couldn't tell from what was presented in methods or SI. The exposure ratio equation seems intuitive enough, but which host data went into this, and how the methods/results were discussed was confusing to me. The exposure ratio equation is based on density of host behaviors and parasites in the environment, but I don't understand how the density of host behaviors is calculated from camera trap images. Specifically, how is the number of individuals assessed, and how does time figure into this density estimate? Is "total exposures" (i.e., line 266) based on total number of individual animals potentially exposed in location x? Is individual animal number assessed at the per-trigger or per-event (number of sequential triggers on the same individual or group) basis? A single animal grazing there for 15 minutes has a much higher probability of exposure (or exposure to a higher number of parasite larvae) than a single animal grazing for 15 seconds. This is part of why I ask several questions below about what individual*seconds means. I don't understand how numbers of individuals and the time they spend was incorporated throughout, and this is an important connection. Additional detail in the methods/results to explain this would be helpful.

Further, this relative exposure ratio is used for analyses whose results are then described as giving a "an estimate of total number of transmissions," "the total number of exposures," "the ratio of total parasite exposures" and "relative exposure". I don't understand how this ratio can give a total number, and how an exposure or transmission can be quantified in such a way (is this a per-host or per-parasite transmission event?). I assume this means the potential number of hosts exposed of those available by species based on their behavior (under various estimates of parasite mortality), but the units here aren't defined. As a relative ratio of potential exposure between site types, this all makes complete sense to me, and Tables S13 and S14 make it clear that this is what is meant by exposure. But the language in the main text is not precise, creating confusion about what this means (in methods, results, and Figure 2F).

192. On a first read through, I thought daily individual*seconds was the interaction of two

variables, the number of individuals recorded in a day and the number of total seconds each spent in activity x? I spent some time scratching my head wondering why you'd multiply the number of individuals by the time spent in the activity, and not sum the total time across all individuals. On a second read through, I figure this has to be connected to the density estimate, but I still don't see how time relates to the number of individuals. Please clarify.

This question came up again for the sections starting at 226. Is total activity based on number of individuals, time, or a combination of individuals and their time spent there? Are visitations coded by individuals (who might trigger the camera several times) or by number of triggers x individuals present? For total, grazing and drinking activity, is the unit of measure time (s/day)? Number of individuals/day? I didn't see additional information to clarify this in the SI.

226 (and throughout this section). The subheading "total transmissions" seems inaccurate. This section is providing an relative index of potential exposures, not a total number of transmission events.

266. This section is similarly using the equation on page 8 to assess relative differences in potential exposure risk. Similar to section 226, I don't think total transmissions or total exposures is an accurate description of this work. (Why is the same analysis called something different under the experimental versus observational systems?).

Related to the potential impact of this paper for understanding heterogeneity of exposure (and not just parasite shedding) in the environment: I don't see in the results section that you consider how the rainfall gradient affects parasite egg counts from soil (something like Figure 4, but for soil parasite data comparing 0m dry to matrix soil parasite estimates along the gradient). This could be a useful way to consider the importance of parasite mortality near water in modulating host exposure risk? If you see no difference between paired 0m vs matrix parasite soil counts along that gradient, you can assume that parasite mortality is probably not that different between dry soil near water and at matrix sites (since environmental differences along that gradient are probably higher than between paired water/matrix sites). If there's a strong effect along the gradient, where matrix sites remain relatively stable (or decline) but near-water sites decline with increasing aridity (or decline significantly more), this would suggest that parasite mortality could be elevated at exposed sites near water. This would help interpretation of the relative exposure results in Figure 2F, and whether we should expect parasite mortality near water to be on the lower or higher side, and hence, how much host aggregation in turn leads to heightened transmission at watering points.

I like Figure 2 very much, walking us through the steps to get from eggs in feces to exposure of the host. (Can you acknowledge the artist who made the buffalo drawing in Figure 2?) Also, can you note in the legend that the letters there show how each sub plot follows the steps of the transmission pathway? I didn't make that connection until a second read through, and it's worth highlighting.

276-280. The results section begins with a summary paragraph, but only describes results for Q1. This could be expanded, if intending to summarize key findings, or else put inside the heading for Q1.

Results for Q2 doesn't refer to Tables S13 and S14. Adding this would be helpful.

Tables 1 and 2 could use more descriptive titles to remind us of the questions being address by these models. What are the response variables being tested for all and individually by species? Table 2 seems to be about dung density from the in-text references, but Table 1 is not referred to in the text, so I can't figure it out. Is this host density from camera trap data?

325 notes that exposure varies by parasite type (parasites of elephants or buffalos). How is this known?

Figures S7 and S8 are too small to read, even zoomed in considerably. These need to be larger to be useful.

As an aside, this paper has parallels to Turner et al 2016 in Scientific Reports, which compares anthrax exposure risk through grazing and drinking behaviors. If the authors aren't aware of this, they may find it interesting.

Minor corrections:

94. replace "raccoon pathogens" with "raccoon parasites"

96 and elsewhere. Why is "parasite sharing" rather than "parasite transmission" the term used to describe cross-species transmission?

101-102. check scientific names (capitals, italics). *Equus burchelli* has been revised to *Equus quagga* (noted here and in SI)

126. remove "results"

259. correct "dung density dung"

469-474: this sentence is long and hard to parse. Perhaps break this into two?

Figure 4. Clarify in the legend that this is parasite density from dung.

Reviewer #1 (Remarks to the Author):

This is a very interesting study that makes important contributions relevant for basic and applied research in disease ecology and conservation biology, and in addition, could be relevant for veterinary and public health. The results clearly demonstrate increased potential for the transmission of gastrointestinal parasites close to permanent water sources in dry areas. A central claim of the paper is that this increased transmission makes these water sources to hotspots of disease exposure. Unfortunately, I am not entirely convinced that the conducted analyses are sufficient for making this claim. However, I think the authors should be able to extend their analyses to fix this problem.

We thank the reviewer for their enthusiasm for our study, and for also raising the important point and suggestion for additional analyses. We have followed this advice, and we now include additional work that aims to scale up our relativized exposure risk estimates across the landscape.

To assess whether water sources act as exposure hotspots the authors compare data collected directly at water sources and data collected at matrix sites with substantial distance to permanent water sources. In their analysis the authors correctly consider variation in parasites (including potential variation in their viability) and variation in individual grazing behaviour and the total time animals spend at either kind of site. A critical issue that seems to be ignored is the landscape-level distribution of areas close to water vs. areas further away from water. Specifically, in savanna ecosystems, areas close to water are usually only a small proportion of the whole landscape. Thus, even if there is increased exposure close to water, animals might spend so little time there that areas close to water still do not act as transmission hotspots. As an extreme example, if the exposure risk close to water is 10 times higher but animals spend only 1% of their grazing time in such areas, then water sources would not really act as exposure hotspots. Or at least transmission close to water would not be of key importance for the overall transmission dynamics. Therefore, I think it would be very valuable if the authors would extend their analyses to estimate how much of the overall exposure across the whole landscape occurs close to water. I realize that the authors do not have the ideal data set to perform such estimations. Nevertheless, some rough estimates should be possible, which I think would be very valuable for judging the ecological relevance of parasite transmission close to permanent water sources.

We thank the reviewer for raising this point. We have now conducted this additional recommended analysis. Since our comparisons were made on the per-unit-area-and-time scale, we weighted our relative potential exposure results by the proportion of land near water (<150m from a provisional water source) compared to the matrix. The 150 meter distance was chosen to parallel the scale of our study. While we did see a notable drop-off in dung and parasite density at the 150m mark, we noted that density was still higher near water relative to matrix sites, making this a conservative estimate of water impacted exposure area.

Specifically, we drew polygons around each watering hole or water pan visible from Google Earth imagery or available from a list provided by ranch owners (including those that were not investigated in the study). This is likely an incomplete set of all water sources as some water sources are small, ephemeral or shaded by canopy, but this

provides a conservative estimate of land impacted by water. We then used the ArcGIS buffer tool to calculate the total area of all 150m rings around each of these polygons (not including the central polygon where water is found). We divided the sum of this area by the total area of each property to determine the percentage of landcover found within 150m of a non-riparian water source. We found that this 150m zone accounted for approximately 1.54% of the total area at Mpala, and 2.61% of the area at OPC. To be conservative, we did not include rivers or streams in this calculation, as animal aggregations in these areas likely differ from the water sources investigated here. However it is highly likely that there is also some degree of aggregation around these latter water sources as well.

Therefore, in scaling up our analyses to the landscape scale (assuming equal mortality near water versus >150m away from it), we found that for cattle at OI Pejeta, for instance, the area near water accounts for approximately 4 times as many total exposures as matrix areas, despite the fact that only ~2.5% of the land is represented.

This translates to approximately 80% of parasite exposures occurring within the area near water, even assuming no aggregation near rivers.

This comes with several caveats, including the assumption described above that our measurement of matrix areas is an accurate representation of all landscape >150m from water. Based on our data it seems likely that there is additional aggregation outside the 150m zone that we did not measure. Conversely, this assumption also misses any transmission that occurs at additional landscape hotspots that might be relevant; e.g. glades and bomas, where animals also aggregate as well as aggregation at other types of water.

We have added text to describe these new methods (L262-267), results (L367-377), and discussion (L483-490), where we emphasize assumptions made in order to arrive at these estimates. Additionally, we have added text describing these methods in the supplement (SI Appendix Section 7), a supplementary figure (SI Appendix Fig S11) to show the map of water sources and buffer zones, and a full results table (SI Appendix Table S15).

In the discussion, the authors briefly discuss the potential for parasite transmission among species. It seems to me there is still some unused potential to explore this topic quantitatively using the collected data set. It might be possible to compare between-species transmission patterns (maybe in form of networks) between water and matrix sites. Such an analysis might reveal that parasite exposure between these sites does not only differ quantitatively (as already shown) but also to some degree qualitatively (in terms of who transmits to whom). I think this could be an interesting addition to the already performed analyses. However, I do not insist to include such an analysis in the current manuscript.

We wholeheartedly agree with this suggestion, and think this is an important consideration. However, between-species transmission is difficult to measure as parasites are cryptic, and developing and analyzing reliable parasite sharing networks require analytical and molecular methods that are beyond the scope of this particular analysis. We are actively working on this question but we believe the addition of this information would make the manuscript difficult to understand and add considerable length to the paper. We hope to follow up on this manuscript with additional analyses

that can expand on the conclusions reviewed here using detailed parasite sharing networks in the future.

Minor comments:

I found the description of the aims in the last paragraph of the introduction not very clear, and more detailed explanations might help a lot. When reading this first I encountered some difficulties:

- There seems to be a contradiction between (1) the first questions which focuses on the quantitative effects of water and (2) the subsequent expectation that focuses on the qualitative effects of experimental water manipulation.

We are not certain if we fully understand this comment, however we have adjusted these first two questions to unify our hypotheses and expectations. While we note that our results do answer the question of ‘to what extent’, we did not have firm *apriori* estimates of this effect size aside from an expectation that is would be highest for the most water-dependent herbivores (L111-112, 115-116, 117).

- For the second question initially, I did not understand the idea of the mortality scenarios. Here it would really help to spell out that you were able to measure this and instead assumed several theoretical scenarios in order to obtain exposure estimates.

We have now added “theoretical parasite mortality scenarios” (L118) to better communicate that potential (rather than measured) mortality was used.

The description of the analyses in the methods appeared to me partly not very well structured. At least I had sometimes difficulties imagining where each specific analysis is fitting into the bigger picture. Here it might be helpful to add information to which question each analysis belongs to. Alternatively, the methods might be easier to follow if the main structure would follow the questions instead of having the main division between experimental and observation parts.

We agree that our organization of the methods could be improved. We created new subheadings that more clearly placed each set of analyses within the scope of the relevant question posed in the introduction, as suggested (L197, 232, 269; L292-298 moved to L254-261).

The section on “total transmission” (L226 ff) could benefit from a more detailed description of what you are doing here. Specifically, it would be useful to explain what “total parasite exposures for each herbivore species” means. As far as I understand it, this combines the exposure of individuals and the density of individuals. I first thought there is something wrong because I had expected that exposure should be analysed on the level of individuals. Only after some time, I realized that this is all correct because with the density of individuals you control for the effect that individuals aggregate at water and thus have a higher exposure risk there. It would be nice if this would be spelt out directly.

We agree that this analysis is a little tricky to explain. Therefore, we have added text to more clearly state that exposure ratio represents the ratio of total exposures (of all individuals summed together) per unit area and time in the region near water, versus the region far from water (L236-239). Additionally, for greater clarity, we provide a dummy example in the supplement to demonstrate how herbivore activity is calculated (SI

Appendix Section 1, under ‘Data Aggregation’).

Throughout the results, statistically non-significant effects should not be reported as indications of the absence of an effect, e.g. as done in lines 310 and 356.

We have endeavored to be more careful with our communication of statistically non-significant results. (L326, L333-334, L383-385, L396).

L388: is it really true that elephants and cattle are much more water-dependent than buffaloes? I thought their water dependence was comparable, but maybe I misremembered that.

It is true that buffalo are highly water dependent. Therefore, we have amended this sentence to say ‘several other animals in our study’. We also note that buffalo are also water-dependent in L439-440. While buffalo are water dependent, they do have additional thermoregulation strategies that cattle do not. For example, buffalo tend to rest during the hottest part of the day and drink water at night, while cattle tend to be herded to water during the hottest noon-time periods, while they rest in cattle corrals at night.

Figure 2: I generally like this figure, I especially like the representation of the panels in the drawing in the centre. I still suggest emphasizing more clearly that the data in panel D do not depict real measurements and instead are assumptions that you put into the analysis. Maybe you could change the text to “Theoretical scenarios for the effect of..”

Thank you for pointing this out. We have now edited the figure to state that this represents theoretical scenarios that were used in the models. We have also added ‘theoretical’ to the figure legend.

Reviewer #2 (Remarks to the Author):

I reviewed a previous version of this manuscript for a different journal. I made several suggestions that I am happy to see the authors addressed here by adding the second main goal, to investigate “to what extent are water sources parasite exposure hot spots across herbivores.” Making the connection from numbers of parasite propagules in the environment to how host exposure risk varies based on heterogeneity in parasite survival under different environmental contexts is a significant strengthening of this study, and this will be an important contribution to the field. I do think, however, that the analyses for this part need more clarification for this connection to be clear.

We thank the reviewer for kindly agreeing to review our paper again! We were very grateful for the very helpful comments provided in our other submission.

What exactly is an “exposure” as defined in the data extraction/analysis? I struggled to understand this throughout, and couldn’t tell from what was presented in methods or SI. The exposure ratio equation seems intuitive enough, but which host data went into this, and how the methods/results were discussed was confusing to me. The exposure ratio equation is based on density of host behaviors and parasites in the environment, but I don’t understand how the density of host behaviors is calculated from camera trap images. Specifically, how is the number

of individuals assessed, and how does time figure into this density estimate? Is “total exposures’ (i.e., line 266) based on total number of individual animals potentially exposed in location x? Is individual animal number assessed at the per-trigger or per-event (number of sequential triggers on the same individual or group) basis? A single animal grazing there for 15 minutes has a much higher probability of exposure (or exposure to a higher number of parasite larvae) than a single animal grazing for 15 seconds. This is part of why I ask several questions below about what individual*seconds means. I don’t understand how numbers of individuals and the time they spend was incorporated throughout, and this is an important connection. Additional detail in the methods/results to explain this would be helpful.

The reviewer raises an important point, and we are grateful to have the opportunity to better explain our calculations. Specifically, since unique individuals could not be identified due to similar markings, we took a ‘patch’ perspective; that is, the density of grazing behaviors in units of individuals x seconds (per unit area over a defined period of time). We realize that this unit of ‘individual*secs’ as written is confusing. It is intended to be similar to ‘person-hours’, to represent the accrual of parasite exposure risk for across all individuals with increasing animal density and time spent in a given area over a defined period of time. Therefore, we have adjusted the term to ‘individual-secs’ to mimic the common ‘person-hours’ unit.

To better explain how individual-secs was calculated, we added text to the methods and supplement, including a dummy example illustrating the process of calculating this unit (L171-175, SI Appendix Section 1, under ‘Data Aggregation’). Specifically, since camera trap images were taken in bursts (at least 3 images with continual triggering if the animal continued to move), we used image timestamps to calculate the duration of animal behaviors.

Further, this relative exposure ratio is used for analyses whose results are then described as giving a “an estimate of total number of transmissions,” “the total number of exposures,” “the ratio of total parasite exposures” and “relative exposure”. I don’t understand how this ratio can give a total number, and how an exposure or transmission can be quantified in such a way (is this a per-host or per-parasite transmission event?). I assume this means the potential number of hosts exposed of those available by species based on their behavior (under various estimates of parasite mortality), but the units here aren’t defined. As a relative ratio of potential exposure between site types, this all makes complete sense to me, and Tables S13 and S14 make it clear that this is what is meant by exposure. But the language in the main text is not precise, creating confusion about what this means (in methods, results, and Figure 2F).

We thank the reviewer for raising this important point. We have endeavored to be more consistent and accurate with our language, and we have edited the methods, results, and figures to state ‘potential parasite exposures’ and ‘potential exposure risk’ when referring to our calculations throughout the manuscript (e.g. L233, 258, 351).

192. On a first read through, I thought daily individual*seconds was the interaction of two variables, the number of individuals recorded in a day and the number of total seconds each spent in activity x? I spent some time scratching my head wondering why you’d multiply the number of individuals by the time spent in the activity, and not sum the total time across all

individuals. On a second read through, I figure this has to be connected to the density estimate, but I still don't see how time relates to the number of individuals. Please clarify.

We realize that this terminology is confusing, especially without further details. To clarify, we multiplied the mean number of individuals observed in a 'trigger set' by the duration of that trigger set to integrate total activity over time. We then summed the total activity for each species within a day to arrive at an estimate of individual-seconds per day.

This question came up again for the sections starting at 226. Is total activity based on number of individuals, time, or a combination of individuals and their time spent there? Are visitations coded by individuals (who might trigger the camera several times) or by number of triggers x individuals present? For total, grazing and drinking activity, is the unit of measure time (s/day)? Number of individuals/day? I didn't see additional information to clarify this in the SI.

We agree that our explanation was unclear. Total activity is based on the combination of individuals and time spent within a given area. Zooniverse users and researchers counted the number of individuals that were present within a given burst of images, and these were later brought together to form a 'trigger set' if images of the same species were taken less than five minutes apart. In essence, activity is the combination of the two units mentioned in the comment – the product of time (s) and number of individuals, summed within a day. As mentioned above, we have endeavored to be more clear in text and now provide a dummy example in the supplement to better illustrate this calculation (SI Appendix Section 1).

226 (and throughout this section). The subheading "total transmissions" seems inaccurate. This section is providing an relative index of potential exposures, not a total number of transmission events.

Thank you for pointing out this inaccuracy. We have now corrected the text to state 'potential exposures'.

266. This section is similarly using the equation on page 8 to assess relative differences in potential exposure risk. Similar to section 226, I don't think total transmissions or total exposures is an accurate description of this work. (Why is the same analysis called something different under the experimental versus observational systems?).

We apologize for our inconsistent use of terminology. Our use of "total transmissions" was a mistake that was not corrected. We have now edited the text to note 'total potential exposures' (L233).

Related to the potential impact of this paper for understanding heterogeneity of exposure (and not just parasite shedding) in the environment: I don't see in the results section that you consider how the rainfall gradient affects parasite egg counts from soil (something like Figure 4, but for soil parasite data comparing 0m dry to matrix soil parasite estimates along the gradient). This could be a useful way to consider the importance of parasite mortality near water in modulating host exposure risk? If you see no difference between paired 0m vs matrix parasite soil counts along that gradient, you can assume that parasite mortality is probably not that

different between dry soil near water and at matrix sites (since environmental differences along that gradient are probably higher than between paired water/matrix sites). If there's a strong effect along the gradient, where matrix sites remain relatively stable (or decline) but near-water sites decline with increasing

aridity (or decline significantly more), this would suggest that parasite mortality could be elevated at exposed sites near water. This would help interpretation of the relative exposure results in Figure 2F, and whether we should expect parasite mortality near water to be on the lower or higher side, and hence, how much host aggregation in turn leads to heightened transmission at watering points.

Thank you very much for this suggestion. We modified our existing model to include the interaction between sample type and annual rainfall. While we found that egg counts tended to be higher in wetter areas, we noted that this pattern was consistent at both watering sources and matrix sites. We have now added these details to the methods (L290), results (L414-417), in addition to providing an accompanying figure in the supplement (SI Appendix Fig S10).

I like Figure 2 very much, walking us through the steps to get from eggs in feces to exposure of the host. (Can you acknowledge the artist who made the buffalo drawing in Figure 2?) Also, can you note in the legend that the letters there show how each sub plot follows the steps of the transmission pathway? I didn't make that connection until a second read through, and it's worth highlighting.

Thank you! We have updated the legend to acknowledge the artist and to better connect the central panel with those surrounding it. The artist is the lead author (G. Titcomb), but this is now explicitly acknowledged in text.

276-280. The results section begins with a summary paragraph, but only describes results for Q1. This could be expanded, if intending to summarize key findings, or else put inside the heading for Q1.

We thank the reviewer for pointing this out. We have now added text to expand upon Q2 key findings, which were missing from this paragraph (L306-308).

Results for Q2 doesn't refer to Tables S13 and S14. Adding this would be helpful.

This is a great suggestion; we have now added references to these tables in L308 and L368.

Tables 1 and 2 could use more descriptive titles to remind us of the questions being address by these models. What are the response variables being tested for all and individually by species? Table 2 seems to be about dung density from the in-text references, but Table 1 is not referred to in the text, so I can't figure it out. Is this host density from camera trap data?

Thank you for pointing this out. We have now provided more informative table captions (L815, L821). Table 1 was incorrectly referred to as ‘Table 2’ in the Q1 section of the results. This has now been updated.

325 notes that exposure varies by parasite type (parasites of elephants or buffalos). How is this known?

We realize that our phrasing is confusing and not completely accurate here. By stating ‘parasites of buffalos and elephants’, we meant to state that relative parasite exposures near water for elephants and buffalos were highest. We have updated our phrasing in L355-356.

Figures S7 and S8 are too small to read, even zoomed in considerably. These need to be larger to be useful.

We have rotated the pages in the supplement and increased the size and quality of these figures for readability, as suggested (SI Appendix Fig S7 and S8).

As an aside, this paper has parallels to Turner et al 2016 in Scientific Reports, which compares anthrax exposure risk through grazing and drinking behaviors. If the authors aren’t aware of this, they may find it interesting.

We thank the reviewer for this suggestion, and we now cite the paper in L.493

Minor corrections:

94. replace “raccoon pathogens” with “raccoon parasites”

We have made this correction as suggested (L94).

96 and elsewhere. Why is “parasite sharing” rather than “parasite transmission” the term used to describe cross-species transmission?

We have changed the phrase to ‘parasite transmission’ as suggested. In other locations, we used the term ‘parasite sharing’ (e.g. as in VanderWaal 2014) to imply that parasites were shared among species, rather than individuals of the same species (L97, 109, 504) We now clarify that where appropriate.

101-102. check scientific names (capitals, italics). *Equus burchelli* has been revised to *Equus quagga* (noted here and in SI)

We have updated the scientific names as suggested (L102-103).

126. remove “results”

We have removed 'results' as suggested (L127).

259. correct "dung density dung"

The phrase now reads "dung density and parasites" (L285).

469-474: this sentence is long and hard to parse. Perhaps break this into two?

We have edited the grammar for readability and split the sentence in two, as suggested (L525-529).

Figure 4. Clarify in the legend that this is parasite density from dung.

We have updated the legend for both this figure and for Figure 3, as suggested (L800, L810).

Reviewers' Comments:

Reviewer #1:

Remarks to the Author:

I have no further comments and recommend the publication of the manuscript.

Reviewer #2:

Remarks to the Author:

The authors did a great job revising the manuscript, and addressed all my comments on the previous version(s). The revision is a much stronger paper, and will be an excellent contribution to the field. There are some minor typos that will be picked up in the proofs, and otherwise, it looks great!